# Long-term glycemic variability and the risk of mortality in diabetic patients receiving peritoneal dialysis

**Hanri Afghahi[1,2], Salmir Nasic[2,3], Björn Peters** [1,2]*, **Helena Rydell[4,5], Henrik Hadimeri[1,6], Johan Svensson[3,6]**

**1** Department of Nephrology, Skaraborg Hospital, Skövde, Sweden, **2** Department of Molecular and Clinical Medicine, Institute of Medicine, The Sahlgrenska Academy at University of Gothenburg, Gothenburg, Sweden, **3** Research and Development Center at Skaraborg Hospital, Skövde, Sweden, **4** Division of Renal Medicine, Department of Clinical Sciences Intervention and Technology, Karolinska Institute, Stockholm, Sweden, **5** Department of Internal Medicine, Swedish Renal Registry, Ryhov Regional Hospital, Jönköping, Sweden, **6** Department of Internal Medicine and Clinical Nutrition, Institute of Medicine, The Sahlgrenska Academy at University of Gothenburg, Gothenburg, Sweden

* bjorn.peters@gu.se

## Abstract

### Background

The large amount of glucose in the dialysate used in peritoneal dialysis (PD) likely affects the glycemic control. The aim of this study was to investigate the association between HbA1c variability, as a measure of long-term glycemic variability, and the risk of all-cause mortality in diabetic patients with PD.

### Methods

325 patients with diabetes and ESRD were followed (2008–2018) in the Swedish Renal Registry. Patients were separated in seven groups according to level of HbA1c variability. The group with the lowest variability was denoted the reference. The ratio of the standard deviation (SD) to the mean of HbA1c, HbA1c (SD)/HbA1c (mean), i.e. the coefficient of variation (CV), was defined as HbA1c variability. Hazard ratios (HR) and 95% confidence intervals (CI) were examined using Cox regression analyses.

### Results

During follow-up, 170 (52%) deaths occurred. The highest mortality was among patients with the second highest HbA1c variability, CV≥2.83 [n = 44 of which 68% patients died]. In the multivariate analyses where lowest HbA1c variability (CV≤0.51) was used as the reference group, HbA1c CV 2.83–4.60 (HR 3.15, 95% CI 1.78–5.55; p<0.001) and CV> 4.6 (HR 2.48, 95% CI 1.21–5.11; p = 0.014) were associated with increased risk of death.

### Conclusion

The high risk of all-cause mortality in patients with diabetes and PD increased significantly with elevated HbA1c variability, as measure of long-term glycemic control. This indicates that stable glycemia is associated with an improvement of survival; whereas more severe

**Data Availability Statement:** The raw data used in the current study are restricted in order to protect participant privacy, as required by data protection acts in Sweden. Data can be made accessible by

request for researchers after permission from the Swedish Ethics Review Authority. Data cannot be shared publicly because of data policy regulations at Skaraborg hospital. Data are available from the Skaraborg hospital Institutional Data Access (contact via skas.dso@vgregion.se) for researchers who meet the criteria for access to confidential data.

**Funding:** The authors received no specific funding for this work.

**Competing interests:** The authors have declared that no competing interests exist.

glycemic fluctuations, possibly caused by radical changes in dialysis regimes or peritonitis, are associated with a higher risk of mortality in diabetic patients with PD.

## Introduction

Globally, diabetes mellitus (DM) is one of the main causes of end-stage renal disease (ESRD) [1, 2]. Patients suffering from DM and chronic kidney disease (CKD) with renal replacement therapy have higher mortality rates compared to the general population [3, 4]. Peritoneal dialysis (PD) is an established renal replacement therapy (RRT) in ESRD patients [5, 6]. The results of most studies in diabetic patients with ESRD have suggested that mortality is approximately similar in patients treated with PD compared with those receiving hemodialysis (HD) [7, 8].

In DM with normal kidney function, elevated mean glycated hemoglobin (HbA1c) levels as well as high glycemic variability were identified as risk factors for worse clinical outcomes and increased mortality [9, 10]. Also in patients with ESRD and dialysis treatment, HbA1c is the most useful marker of long-term glycemic control [11, 12], despite the shortened erythrocyte life span in ESRD [13]. Moreover, high level of HbA1c variability, as a measure of glycemic variability, has been associated with increased risk of diabetic microvascular and macrovascular complications as well as all-cause mortality [14, 15].

Earlier studies evaluated the association between the mean value of HbA1c and the risk of cardiovascular events or mortality in patients with DM and PD [16–18]. However, the impact of the glucose load from the PD dialysate on glycemic variability in relation to mortality has previously not been investigated in diabetic patients.

The aim of the present study was to investigate the association between HbA1c variability, a measure of glycemic variability, and the risk of all-cause mortality in diabetic patients with maintenance PD.

## Material and methods

The Regional Ethics Review Board at the University of Gothenburg approved the study (Dnr 698–17, Exp 2017-09-21) that was performed in accordance with the Declaration of Helsinki. In this study all data were fully anonymized before we accessed them. All patients were informed adequately by hospitals or dialysis centers about the Swedish Renal Register (SRR). The patients were completely anonymous.

In this study, 795 patients (age 66±13 years, 71% men) with DM and ESRD receiving PD treatment were followed 2008–2018. The patients with less than three months on PD were excluded from the study (n = 48). Of the remaining patients, 325 had at least two values of HbA1c and were included in the analyses of HbA1c variability. The mean follow-up period was 3.0±3.2 years. HbA1c was 6.8% ± 2.4% on average (median 6.6%, range 4.2%-19.7%). At baseline, 234 (72%) of the patients received treatment with Continuous ambulatory peritoneal dialysis (CAPD) and 91 (28%) of the patients were on Automated Peritoneal Dialysis (APD).

The data were extracted from the Swedish Renal Register (SRR). SRR is a computerized, web-based quality register in Sweden for patients with chronic renal failure. SRR is based on annual cross-sectional surveys from all nephrology departments and dialysis units in Sweden.

### Clinical classification

We defined all patients in the SRR with a diagnosis of DM type 1 or type 2 diabetes or diabetes nephropathy as a patient with DM. Age was defined as the baseline age, whereas the mean

values during follow-up were used in the analyses in terms of body mass index (BMI), blood pressure (BP), and laboratory variables.

In this study at baseline, 292 (90%) of the patients had at least one medication for treatment of high blood pressure. We defined hypertension as systolic blood pressure (SBT) > 140 mmHg or diastolic blood pressure (DBP) > 90 mmHg. According to this definition, 146 (47%) of the patients had hypertension at baseline.

All data of cardiovascular disease (CVD) and malignancy were obtained from the SRR.

## Statistical methods

Baseline clinical and biochemical characteristics are presented as mean values ± standard deviation (SD) or as frequencies and proportions (n, %). For univariate comparisons with respect to continuous variables, the t-test for independent samples was used and for comparisons with respect to categorical variables, the Chi-2 test was used. HbA1c variability was determined as the coefficient of variation (CV) for HbA1c for each patient (the ratio between the standard deviation (SD) of HbA1c and the mean of HbA1c, i.e. HbA1c (SD)/ HbA1c (mean)). To examine the relationship between HbA1c variability and all-cause mortality in univariate analyses we used Kaplan-Meier survival analysis and for further multivariate analysis we performed Cox regression models and hazard ratios (HR) with 95% confidence intervals (CI) were estimated. Proportional hazards assumption was controlled, and it was not violated. The patients were divided in seven groups according to HbA1c-level. The number of groups was defined based on the purpose to include sufficient number of patients in each class and at the same time yield a wideness in CV for HbA1c. The group of patients with the lowest HbA1c variability CV≤0.51 was used as the reference group for calculations of HR and statistical comparisons. Models were adjusted for the all covariates that turned out as statistically significant or when p-value<0.1 in the univariate analysis. Interactions between HbA1c variability and the covariates were tested and were found to be non-significant for all covariates. All statistical analyses were performed by statistical package IBM SPSS v.25 and p-value<0.05 were considered as statistically significant.

## Results

### Clinical and biochemical characteristics

Table 1 presents clinical and biochemical characteristics of the included patients. Mean age was 66 ±14 years, mean HbA1c was 6.8 ± 2.4%, and the mean values of SBP and DBP were 137 ± 17 mmHg and 76 ± 9 mmHg, respectively. The patients had a dialysis vintage of 4.3±4.3 years. The majority of the patients had slightly low serum albumin (31 ± 5 g/L). The patients mostly followed Chronic kidney disease–mineral and bone disorder (CKD–MBD) KGIDO [19] as the mean serum phosphate was 4.8 ± 1.0 mg/dl and the mean PTH was 268 ± 170 pg/mL. Twenty-five percent of the patients had a previous history of CVD and three percent had a previous history of malignancy.

In the analyses of HbA1c CV, there were different number of HbA1c measurements in individual patients (between 2 and 12 measurements), but the number of measurements was associated with the level of HbA1c variability, Table 2.

### All-cause mortality

During the follow-up, 170 (52%) of the patients died. In the subgroup of patients > 75 years old (n = 77), mortality was 70% and in patients ≤ 75 years old (n = 248), mortality was 47%. The patients who died during the study compare to all patients were older 69.5±10.6 years, more men (75%), lower DBP 73±9 mmHg and 32% had previous history of CVD.

**Table 1. Clinical and biochemical characteristics in the study population (n = 325) of diabetic patients receiving PD treatment.**

|  | All patients n = 325 | Dead n = 170 | Alive n = 155 | p-value[1] |
|---|---|---|---|---|
| **Age (years)** | 65.9 ± 13 | 69.5±10.6 | 61.9±13.9 | <0.001 |
| **Men n (%)** | 230 (71%) | 128 (75%) | 102 (66%) | 0.060 |
| **Time since PD-start (years)** | 4.3±4.3 | 4.5±3.5 | 4.0±5.0 | 0.316 |
| **HbA1c %** | 6.8± 2.4 | 6.9±2.5 | 6.8±2.6 | 0.292 |
| **Systolic blood pressure (mmHg)** | 137 ± 17 | 137±19 | 136±15 | 0.666 |
| **Diastolic blood pressure (mmHg)** | 76 ± 9 | 73±9 | 78±9 | <0.001 |
| **MAP** | 96±10 | 95±10 | 97±9 | 0.012 |
| **Antihypertensive treatment, n(%)** | 311 (96%) | 160 (94%) | 151 (97%) | 0.143 |
| **BMI (kg/m$^2$)** | 27 ± 4.6 | 27±5 | 27±4 | 0.952 |
| **Hemoglobin (g/L)** | 118 ± 10 | 117±10 | 119±10 | 0.327 |
| **CRP (mg/mL)** | 11.5 ± 14.8 | 13.5±16.6 | 9.2±12.2 | 0.008 |
| **Serum albumin (g/L)** | 31 ± 4.7 | 29.8±4.3 | 32.3±4.7 | <0.001 |
| **Serum phosphate (mg/dL)** | 4.8 ± 1.0 | 4.8±1.1 | 4.9±1.1 | 0.452 |
| **PTH (pg/mL)** | 268±170 | 267±190 | 280±159 | 0.268 |
| **Total cholesterol (mmol/L)** | 4.5 ± 1.3 | 4.5±1.2 | 4.6±1.3 | 0.480 |
| **Previous history of CVD n(%)** | 83 (25%) | 55 (32%) | 28 (18%) | 0.003 |
| **Previous history of malignancy n(%)** | 10 (3%) | 4 (2%) | 6 (4%) | 0.429 |

Data are presented as means ± SD or frequencies (%). Age was defined as the baseline age, whereas BMI, blood pressure, and biochemical variables for each patient were defined as the average of all measurements during the study period.

[1]p-value consider comparison between dead and alive patients.

**Association between HbA1c variability and the risk of all-cause mortality.** The association between HbA1c variability and the risk of all-cause mortality is shown in Table 3 and Fig 1. The patients with the lowest HbA1c variability (the reference group) had the lowest rate of mortality (CV≤0.51, n = 25, 38% died). The highest incidence of mortality was observed in the group with CV≥2.83 (n = 44, 68%).

In the multivariate analyses, in which adjustments were made for age, sex, MAP, CRP, serum albumin and CVD, the risk of all-cause mortality was significantly increased in the HbA1c variability group with CV 2.83–4.60 (HR 3.15, 95% CI 1.78–5.55, p-value<0.001), and the HbA1c variability group with CV> 4.6 (HR 2.48, 95% CI 1.21–5.11, p-value = 0.014), Table 3.

Median survival time in total population was (HR 3.1, 95% CI 2.7–3.4). The two groups with highest CV analyses had lowest median survival time; CV 2.83–4.60 (HR 1.9, 95% CI 1.6–

**Table 2. Distribution of number of HbA1c measurements per patient according to classes of HbA1c variability.**

|  | Number of measurements per patient % (n) | | |
|---|---|---|---|
| **Coefficient of variation (CV) for HbA1c** | **2 measurements** | **3–4 measurements** | **5–12 measurements** |
| **CV≤ 0.51; n = 65** | 55.4% (36) | 21.5% (14) | 23.1% (15) |
| **0.51 < CV ≤ 0.77; n = 52** | 36.5% (19) | 38.5% (20) | 25.0% (13) |
| **0.77 < CV ≤ 1.32; n = 53** | 32.1% (17) | 39.6% (21) | 28.3% (15) |
| **1.32 < CV ≤ 1.77; n = 35** | 54.3% (19) | 22.9% (8) | 22.9% (8) |
| **1.77 < CV ≤ 2.83; n = 53** | 37.7% (20) | 50.9% (27) | 11.3% (6) |
| **2.83 < CV ≤ 4.60; n = 44** | 75.0% (33) | 22.7% (10) | 2.3% (1) |
| **CV > 4.60; n = 23** | 65.2% (15) | 34.8% (8) | 0% (0) |
| **Total; n = 325** | 48.9% (159) | 33.2% (108) | 17.8% (58) |

**Table 3. Univariate and multivariate analyses of the association between HbA1c variability and the risk of all-cause mortality.**

| HbA1c Coefficient of variation (CV) | Univariate analyses | | Multivariate analyses | |
|---|---|---|---|---|
| | HR (95% CI) | p-value | HR (95% CI) | p-value |
| CV≤ 0.51; n = 65 | reference | | reference | |
| 0.51 < CV ≤ 0.77; n = 52 | 1.23 (0.69–2.20) | 0.474 | 1.12 (0.62–2.01) | 0.701 |
| 0.77 < CV ≤ 1.32; n = 53 | 1.42 (0.83–2.41) | 0.197 | 1.29 (0.75–2.20) | 0.350 |
| 1.32 < CV ≤ 1.77; n = 35 | 1.62 (0.89–2.95) | 0.117 | 1.35 (0.73–2.49) | 0.336 |
| 1.77 < CV ≤ 2.83; n = 53 | 1.50 (0.88–2.56) | 0.132 | 1.37(0.80–2.35) | 0.250 |
| 2.83 < CV ≤ 4.60; n = 44 | 3.66 (2.11–6.33) | <0.001 | 3.15 (1.78–5.55) | <0.001 |
| CV > 4.60; n = 23 | 3.09 (1.56–6.12) | 0.001 | 2.48 (1.21–5.11) | 0.014 |

Hazard ratios (HR) with 95% confidence intervals (CI) were calculated using Cox regression analyses. In the multivariate analyses, adjustments were made for the all variables that turned out as statistically significant in Table 1.

2.2) and CV> 4.60 (HR 2.0, 95% CI 1.4–2.6). The group of patients with lowest CV≤0.5 had highest median survival time (HR 4.3, 95% CI 2.6–5.9), p-value<0.001 (Table 4).

Free survival rates between the seven subgroups during the followed-up time by Kaplan–Meier survival curves is shown in Fig 1. The mean survival time was reduced in groups with higher CV of HbA1c. The association between HbA1c variability and all-cause mortality is visualized in Fig 2. Thus, in both the unadjusted and adjusted models, the HR of death increased markedly i with higher HbA1c variability.

## Discussion

In this study, 795 diabetic patients with ESRD and PD treatment were followed for a mean of 3.0± 3.2 years. Of these patients, the 325 who had at least two measurements of HbA1c were

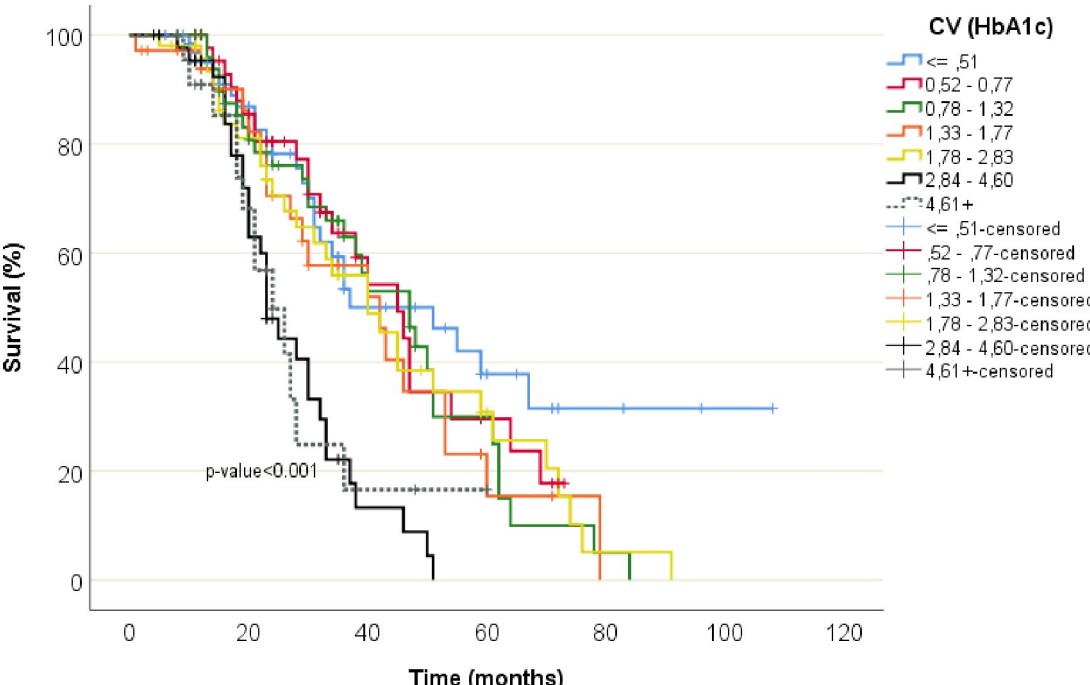

**Fig 1. All-cause survival (Kaplan-Meier curves) according to HbA1c variability.** Kaplan–Meier curves showing the survival reduced with increase of HbA1c variability.

**Table 4. Survival data according to HbA1c variability.**

| Coefficient of variation (CV) for HbA1c | Nr of patients at risk | Mortality n (%) | Median survival time (years) with 95% CI |
|---|---|---|---|
| CV ≤ 0.51 | 65 | 25 (38%) | 4.3 (2.6–5.9) |
| 0.51 < CV ≤ 0.77 | 52 | 22 (42%) | 3.8 (2.9–4.6) |
| 0.77 < CV ≤ 1.32 | 53 | 31 (58%) | 3.9 (3.0–4.8) |
| 1.32 < CV ≤ 1.77 | 35 | 19 (54%) | 3.5 (2.2–4.8) |
| 1.77 < CV ≤ 2.83 | 53 | 30 (57%) | 3.3 (2.5–4.2) |
| 2.83 < CV ≤ 4.60 | 44 | 30 (68%) | 1.9 (1.6–2.2)* |
| CV > 4.60 | 23 | 13 (56%) | 2.0 (1.4–2.6)* |
| Total | 325 | 170 (52%) | 3.1 (2.7–3.4) |

*Statistically significant compared in pairwise comparisons vs all other classes (CV<2.83).

included in the analyses of HbA1c variability. 49% of the patients had 2 measurements and 51% had between 3 and 12 measurements. We showed that high HbA1c variability, as a measure of long-term glycemic variability, was associated with a markedly increased risk of all-cause mortality in diabetic patients with ESRD and PD treatment.

The mean age of our population was 66 years, which is almost similar to the general population with ESRD and dialysis treatment. The mean HbA1c value was 6.8%, which indicates that most of the diabetic patients had adequate long-term glycemic control. Furthermore, the patients mostly had acceptable BP during the study (SBP 137 ±17 mmHg, DBP 76 ±9mmHg). The mean serum albumin was also under adequate control (31±5 g/L). As a result, our population was mostly relevant in terms of the aims of the present study.

Previous studies have assessed the relationship between the mean HbA1c level and the risk of CVD or mortality in diabetic patients with ESRD and dialysis [20, 21]. The results are

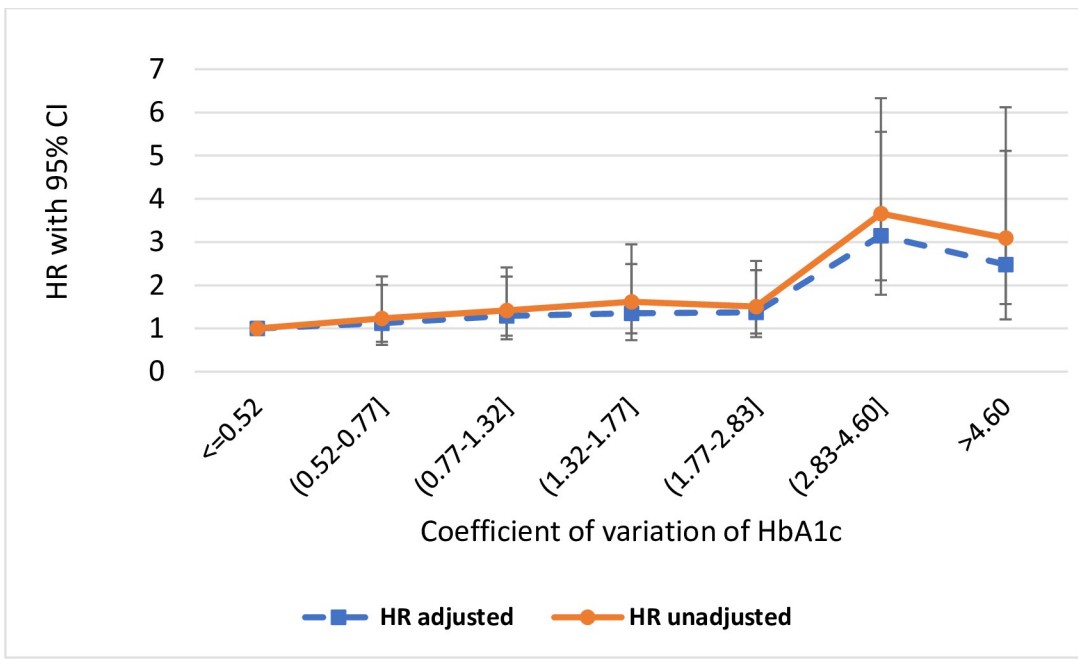

**Fig 2. Hazard ratio (HR) of all-cause mortality according to variability of HbA1c levels.** The association between HbA1c variability and the risk of all-cause mortality. Hazard ratios (HR) and 95% confidence intervals (CI) based on univariate and multivariate model are presented.

diverse, but in general, HbA1c>8% was associated with higher mortality risk [22]. Moreover, in several studies, HbA1c<6% was associated with elevated risk of mortality in chronic dialysis [20, 23]. Based on these findings, it has been suggested that there is a J- or U-shaped relationship between the absolute HbA1c level and the risk of mortality [11, 14]. However, in the present study of diabetic patients with ESRD and PD treatment, only high HbA1c variability was associated with increased risk of all-cause mortality. In patients with DM and preserved renal function, increased glycemic variability, which suggests more severe glycemic fluctuations, has been associated with increased risk of cardiovascular events [12, 15]. Furthermore, previous studies have indicated that fluctuations in glycemic control could cause more damage to endothelial cells than a persistently high level of glucose [24, 25]. Underlying mechanisms could be that glucose fluctuations lead to oxidative stress and impaired endothelial function [26]. Elevated glucose variability also increases the occurrence of hypoglycemia, which is associated with a higher risk of mortality among patients with DM [27–29].

However, little is known of the importance of glycemic variability in patients with CKD.

Most previous studies have included the patients with mild to moderate renal dysfunction. The results of a study by Lee et al. showed that greater HbA1c variability was associated with an increased risk of cardiovascular events in type 2 DM with preserved renal function, but not in moderate to advanced CKD. This study included 1834 patients with eGFR < 60 min/ml/1.73m2 (mean eGFR 39.9 ± 14.8 min/ml/1.73m2; follow-up 6.3 years), but the patients that received dialysis were not examined separately [30]. In stages 3–4 of CKD, greater HbA1c variability was associated with a decreased risk of progression to dialysis, but this association was not found in CKD stage 5. Although patients receiving PD treatment were not specifically analyzed, this could suggest that in diabetic patients, aggressive glycemic control in CKD stages 3–4 is most important in terms of the clinical outcomes [31].

The results of the Dialysis Outcomes and Practice Patterns (DOPPS) study suggested that higher HbA1c values than those currently recommended for a dialysis population were associated with less variability of glucose and a decreased risk of mortality in patients with DM and HD [14].

In diabetic patients receiving PD, a large amount of glucose is absorbed from the dialysate, which obviously could increase glucose variability as compared with a HD population. Additionally, the impact of changes in the dialysis regime by the use of lower or higher glucose concentration in the dialysate as well as the importance of peritonitis and inflammation on glucose fluctuations in PD is unclear. To our knowledge, the present study is the first to evaluate the association between HbA1c variability, as a measure of long-term glycemic fluctuations, and the risk of all-cause mortality in diabetic patients with PD.

In the present study, we used HbA1c as a measure of long-term glycemic control, but the validity of this marker in patients with ESRD and dialysis is debated [13]. The use of HbA1c in populations with ESRD and dialysis has several limitations since the shortened erythrocyte life span and metabolic acidosis can affect the HbA1c level [32]. Moreover, it has been suggested that glycated albumin and fructosamine could be better markers of long-term glycemic control than HbA1c in a dialysis population [33, 34]. However, the level of glycated albumin is highly affected by inflammation and low serum albumin, which is common in PD [32]. Therefore, as in the present study, HbA1c is generally used to evaluate long-term glycemic control in ESRD patients receiving dialysis treatment.

The major strengths of this cohort study are the nationwide scale and the large number of patients and events, which likely resulted in adequate statistical power and high external validity. Furthermore, the majority of our patients were followed according to national guidelines for PD treatment. Thus, the study population was mostly homogeneous, which suggests high internal validity. Furthermore, high BMI and obesity are important risk factors for mortality

among PD patients [35, 36]. Our patients tended to be slightly overweight (BMI 27 ± 4.6 kg/m2), but in the multivariate analyses, the association between high HbA1c variability and increased risk of all-cause mortality remained significant also after adjustment for BMI and multiple other covariates. Finally, the high incidence of mortality (52% of the patients died) is consistent with the fact that 24% of our patients was more than 75 years old.

The present study has some limitations. In our observational study, a cause–effect relationship cannot be established. The data on dialysis effectivity (Kt/v), type of peritoneum ("slow transporters", "fast transporters") and dialysis regime were not recorded, and especially the influence of the use of icodextrin or non-glucose based peritoneal dialysis solution (Nutrineal) on glycemic variability would have been of interest to delineate [37–41]. Furthermore, data on peritonitis as a complication of PD, which can affect ultrafiltration and glucose absorption [42], were not available in the present study. All the included patients were treated with insulin, but the daily insulin doses were not available. Finally, of the totally 795 patients with DM and ESRD receiving PD treatment, we could include the 325 patients with at least three months on PD and two or more values of HbA1c.

In conclusion, to our knowledge, the present study is the first to evaluate the association between glycemic variability and the risk of all-cause mortality in diabetic patients receiving PD treatment. In this observational study using data from the SRR of diabetic patients with maintenance PD, high HbA1c variability, as a measure of long-term glycemic control, was significantly associated with increased risk of all-cause mortality. Therefore, higher magnitudes of glycemic fluctuations, which might be caused by radical changes in dialysis regimes or peritonitis, are associated with higher risk of mortality in this group of patients. Further studies are needed to evaluate whether reduced glycemic variability by improved clinical care can reduce the high mortality seen in patients with DM and PD.

## Acknowledgments

We would like to thank all the patients, the SRR and all participating nurses, physicians, and other staff in Sweden. Most of all, we would like to thank the staff at the Research and Development Center (FoU) at Skaraborg Hospital, Skövde, Sweden for important technical assistance in this study.

## Author Contributions

**Conceptualization:** Hanri Afghahi, Salmir Nasic.

**Data curation:** Hanri Afghahi, Salmir Nasic, Björn Peters, Helena Rydell.

**Formal analysis:** Hanri Afghahi, Salmir Nasic, Björn Peters.

**Investigation:** Hanri Afghahi, Salmir Nasic.

**Methodology:** Hanri Afghahi, Salmir Nasic, Björn Peters, Johan Svensson.

**Project administration:** Hanri Afghahi.

**Resources:** Hanri Afghahi, Helena Rydell.

**Supervision:** Hanri Afghahi, Björn Peters.

**Validation:** Hanri Afghahi, Salmir Nasic, Björn Peters, Helena Rydell, Henrik Hadimeri, Johan Svensson.

**Visualization:** Hanri Afghahi.

**Writing – original draft:** Hanri Afghahi.

**Writing – review & editing:** Salmir Nasic, Björn Peters, Helena Rydell, Henrik Hadimeri, Johan Svensson.

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
