## [Decision Letter · Decision Letter 0]

14 Jul 2021

PONE-D-21-18957

LONG-TERM GLYCEMIC VARIABLITY AND THE RISK OF MORTALITY IN DIABETIC PATIENTS RECEIVING PERITONEAL DIALYSIS

PLOS ONE

Dear Dr. Peters,

Thank you for submitting your manuscript to PLOS ONE. After careful consideration, we feel that it has merit but does not fully meet PLOS ONE’s publication criteria as it currently stands. Therefore, we invite you to submit a revised version of the manuscript that addresses the points raised during the review process.

Please address all the issues raised by the reviewer. If you do not have the clinical data necessary to make the association that you have proposed in the title robust and consistent with clinical reality, you may shorten the article with a statistical description of what you find from the data available. 

We look forward to receiving your revised manuscript.

Kind regards,

Vivekanand Jha

Academic Editor

PLOS ONE

Journal Requirements:

2. In your ethics statement in the Methods section and in the online submission form, please provide additional information about the data used in your retrospective study. Specifically, please ensure that you have discussed whether all data were fully anonymized before you accessed them and/or whether the IRB or ethics committee waived the requirement for informed consent. If patients provided informed written consent to have data from their medical records used in research, please include this information.

4. We noticed you have some minor occurrence of overlapping text with the following previous publication, which needs to be addressed:

- https://academic.oup.com/ndt/article/34/Supplement_1/gfz101.SaO059/5515881

In your revision ensure you cite all your sources (including your own works), and quote or rephrase any duplicated text outside the methods section. Further consideration is dependent on these concerns being addressed.

Reviewers' comments:

Reviewer's Responses to Questions

**Comments to the Author**

1. Is the manuscript technically sound, and do the data support the conclusions?

Reviewer #1: Yes

Reviewer #2: Partly

2. Has the statistical analysis been performed appropriately and rigorously? 

Reviewer #1: N/A

Reviewer #2: N/A

3. Have the authors made all data underlying the findings in their manuscript fully available?

Reviewer #1: No

Reviewer #2: Yes

4. Is the manuscript presented in an intelligible fashion and written in standard English?

Reviewer #1: Yes

Reviewer #2: Yes

5. Review Comments to the Author

Reviewer #1: The authors have studied the long term glycemic variability and risk of mortality in diabetics on peritoneal dialysis (PD), which is a novel and good concept. May I comment the following:

1. As the title mentions 'long term variability', it is preferable to have a follow-up of at least one year on PD with four HbA1c readings. In the manuscript, follow-up of 3.0+/3.2 years has been mentioned, which means that some patients had a very short follow-up and some patients had only two HbA1c readings. Also, since data is not normally distributed, better to present it as median and range

2. Kindly mention the PD regimes in the population studied and adequacy of PD (if possible)

3. Separating the patients in standard five quintiles may be preferable

4. Would like to know the mortality in a similar group of non-diabetics on PD in the Registry

Reviewer #2: 1. This seems to be a retrospective observational study retrieving data from a National (Swedish) Registry. The aim of this study was to investigate the association between HbA1c variability, as a measure of long-term glycemic variability, and the risk of all -cause mortality in diabetic patients with PD.

2. In the first paragraph of the Methods section, the authors state that 795 diabetic PD patients were followed up between 2008-2018, but 48 were excluded as they had been on PD for less than three months. Of the remaining patients, 325 had at least two values of HbA1c and were included in the HbA1c variability analyses with a mean follow-up period of 3.0 � 3.2 years. Would you confirm that 322 were excluded for not having, at least, two values of HbA1c ? Therefore, a total of 370 patients (roughly 46 % of all diabetic PD patients on the Registry) were excluded for either being on PD for less than three months or not having two values or more of HbA1c during the follow-up period (3.0 � 3.2 years).

3. Certainly, all 795 diabetic PD patients followed up between 2008-2018 must have had more than two values of HbA1c analysed during their follow-up period, but most likely not captured for the Registry. Can this be considered a bias ? a limitation of the study ? and what about the variability of the results measured in different laboratories around Sweden ? were all PD patients in Sweden evaluated for their HbA1c levels with the same lab method ?

4. Is it possible to know how many of the 322 patients excluded for not having, at least two values of HbA1c, had no (0) values of HbA1c ?

5. In the Introduction the authors write ……… Earlier studies evaluated the association between the mean value of HbA1c and the risk of cardiovascular events or mortality in patients with DM and PD [16-18]. However, the impact of the glucose load from the PD dialysate on glycemic variability has not been previously investigated in diabetic patients. In order to evaluate the impact of glucose load from the PD fluid on glycemic variability, it is fundamental to have important clinical information which may significantly impact on HbA1c variability , the most important being:

a. Data on which modality (APD or CAPD) and the daily prescription for each patient. Were there switches from APD to CAPD or from CAPD to APD in any of the patients included, during the observation period ?

b. CAPD daily dialysis prescription: number of exchanges, volume per exchange, glucose concentration in each glucose-based PD solution, use of the non-glucose based PD solutions icodextrin (Extraneal) and or aminoacids solution (Nutrineal)

c. APD daily dialysis prescription: number of exchanges, volume per exchange, total volume used, glucose concentration in each glucose-based PD solution, use of the non-glucose based PD solutions icodextrin (Extraneal) and or aminoacids solution (Nutrineal)

d. Daily insulin prescription in case of insulin-dependent diabetics and other drugs for non-insulin dependent diabetics. How many insulin-dependent and how many non-insulin dependent patients were on the study ?

e. How was the peritoneal membrane transport characteristics distribution among the CAPD and APD patients ? High transporters usually absorb more glucose and this can impact on morbidity and mortality.

f. Information on blood pressure (percentage of hypertensive patients) and/or fluid overload status.

g. Use of BRA, ACEi, aspirin, beta-blockers

h. Presence of co-morbidities ( Charlson index ?)

i. Information on peritonitis rates, Kt/V, residual renal function.

6. In the introduction the Authors wrote…. However, the impact of the glucose load from the PD dialysate on glycemic variability has not been previously investigated in diabetic patients. Information above described from a to h should be provided in order to corroborate this assumption.

7. The authors conclude….” In conclusion, to our knowledge, the present study is the first to evaluate the association between glycemic variability and the risk of all-cause mortality in diabetic patients receiving PD treatment. In our diabetic patients with maintenance PD, high HbA1c variability, as a measure of long-term glycemic control, was significantly associated with increased risk of all-cause mortality. Therefore, higher magnitudes of glycemic fluctuations, which might be caused by radical changes in dialysis regimes or peritonitis, are associated with higher risk of mortality in this group of patients. Further studies are needed to evaluate whether improved clinical care can reduce glycemic variability as well as the high mortality seen in patients with DM and PD. “Are they sure that “they are the first to evaluate the association between glycemic variability and the risk of all-cause mortality in diabetic patients receiving PD treatment.” ?

8. When the authors write “In our diabetic patients with maintenance PD…”, isn’t it more appropriate to write “In the SRR diabetic patients group with maintenance PD…” ?

9. I cannot see support to the following sentence “Therefore, higher magnitudes of glycemic fluctuations, which might be caused by radical changes in dialysis regimes or peritonitis, are associated with higher risk of mortality in this group of patients “ as there is no information or data on dialysis prescriptions or peritonitis rates analysed in this study as well as of a definition of the word “radical” in the context of “radical changes in dialysis regimes or peritonitis “. In order to address this association (changes in dialysis prescriptions or peritonitis) with mortality, what is needed to investigate is primarily not the glycemic variability (a consequence of glucose exposure), but the changes in the daily glucose exposure/load and its clinical impact on different sub-groups of diabetic PD patients. In diabetic PD patients, higher HbA1c levels may indicate greater cumulative peritoneal glucose exposure with its attendant damage to the peritoneal membrane

10. The authors wrote as last sentence in the conclusion…. “Further studies are needed to evaluate whether improved clinical care can reduce glycemic variability as well as the high mortality seen in patients with DM and PD. “. I would challenge this conclusion using as basis, two not recent published randomized trials (Paniagua et al, de Moraes et al), already showing improved clinical care by improving metabolic control, decreasing glucose load and exposure as well as optimizing fluid management in PD patients (both diabetics and non-diabetics). I suggest reading the RCT CAPD study by Ramon Paniagua et al. and the RCT APD study by Thyago de Moraes et al.

Paniagua R, Ventura MD, Avila-Diaz M et al. Icodextrin improves metabolic and fluid management in high and high-average transport diabetic patients. Perit Dial Int 2009; 29: 422–432

de Moraes T.P., Andreoli M.C., Canziani M.E. et al. Icodextrin reduces insulin resistance in non-diabetic patients undergoing automated peritoneal dialysis: results of a randomized controlled trial (STARCH). Nephrol Dial Transplant. 2015; 30: 1905-1911

I also suggest reading the RCT Impendia study by Li PK et al

In the IMPENDIA study, the primary endpoint was change in glycated hemoglobin from baseline. Mean glycated hemoglobin at baseline was similar in both groups. During the six months of the study, in the intention-to-treat population, the mean glycated hemoglobin profile improved in the intervention group but remained unchanged in the control group (0.5% difference between groups; 95% confidence interval, 0.1% to 0.8%; P=0.006).

Li PK, Culleton BF, Ariza A et al. Randomized, controlled trial of glucose sparing peritoneal dialysis in diabetic patients. J Am Soc Nephrol 2013; 24: 1889–1900

As well as reading the paper by McIntyre et al on glycemic control in diabetic CAPD patients assessed by CGMS

A practical approach to reduce disturbances of the carbohydrate metabolism in PD patients is the reduction of glucose exposure by also prescribing glucose-sparing solutions. In a study involving eight diabetic CAPD patients, replacement of a glucose-based regimen with a Physioneal-Extraneal-Nutrineal regimen was associated with a reduction in the 24-hour variability of glucose concentrations as measured by a subcutaneous probe in the interstitial fluid of the abdominal wall.

Marshall J, Jennings P, Scott A, Fluck RJ, McIntyre CW: Glycemic control in diabetic CAPD patients assessed by continuous glucose monitoring system (CGMS). Kidney Int 64: 1480–1486, 2003

11. Lastly, it should be taken into account that in type 1 and 2 diabetics (not in dialysis), HbA1c is not a good predictor of cardiovascular disease (CVD), whereas insulin resistance is predictive of CVD and indeed may be the most important single cause of coronary artery disease (see for example: Home P. Diabetes Care 2019; 42:1615-23; Adeva-Andany MM et al. Diabetes Metab Syndr 2019; 13:1449-55; Shahim B. et al. Diabetes Care 2017:40:1233-40; Eddy D. et al. Diabetes Care 2009; 32; 361-66; Orchard TJ et al. 2003; 26:1374-79). In addition, since patients in dialysis are often affected by subclinical/clinical anemia, which reduces red blood cell survival and hence Hb glycosylation, this introduces an additional bias in the interpretation of HbA1c variability. At this regard, it should be kept in mind that the simple evaluation in dialysis patients of total Hb levels would be of little help in figuring out potential differences in the glycosylation rate as total Hb levels. Indeed, the latter (Hb levels) may be easily corrected by Epo treatment, but it doesn’t tell you anything about survival rate of RBCs, one of the major determinant of the extent of HB glycosylation.

In my opinion the paper needs a major revision. If the Authors do not have the clinical data necessary to make the association in the title robust and as a reflection of the clinical reality, you may shorten the article with a statistical description of what you see from the data available. I am not an expert in Statistics, but as a clinician, I consider the clinical data I described above of utmost importance in order to keep the paper in the present format. So I really hope you have the data I am suggesting to add to the paper.

6. PLOS authors have the option to publish the peer review history of their article (what does this mean?). If published, this will include your full peer review and any attached files.

Reviewer #1: No

Reviewer #2: **Yes: **Jose Carolino Divino-Filho

---

## [Author Response · Author response to Decision Letter 0]

26 Sep 2021

Responses to Reviewers comments

Journal Requirements:

Response to comment 1: Yes, we have checked PLOS ONE´s style requirements.

2. In your ethics statement in the Methods section and in the online submission form, please provide additional information about the data used in your retrospective study. Specifically, please ensure that you have discussed whether all data were fully anonymized before you accessed them and/or whether the IRB or ethics committee waived the requirement for informed consent. If patients provided informed written consent to have data from their medical records used in research, please include this information.

Response to comment 2: In this study all data were fully anonymized before we accessed them. All patients were informed adequately by hospitals or dialysis centers about SRR. The patients were completely anonymous.

Response to comment 3: 

The raw data used in the current study are restricted in order to protect participant privacy, as required by data protection acts in Sweden. Data can be made accessible by request for researchers after permission from the Swedish Ethics Review Authority. Data cannot be shared publicly because of data policy regulations at Skaraborg hospital. Data are available from the Skaraborg hospital Institutional Data Access (contact via skas.dso@vgregion.se) for researchers who meet the criteria for access to confidential data.

4. We noticed you have some minor occurrence of overlapping text with the following previous publication, which needs to be addressed:

- https://academic.oup.com/ndt/article/34/Supplement_1/gfz101.SaO059/5515881

In your revision ensure you cite all your sources (including your own works), and quote or rephrase any duplicated text outside the methods section. Further consideration is dependent on these concerns being addressed.

Response to comment 4: The above-mentioned publication was an abstract presented at ERA-EDTA-congress in Budapest/Hungary in 2019.

Reviewer 1

1. As the title mentions 'long term variability', it is preferable to have a follow-up of at least one year on PD with four HbA1c readings. In the manuscript, follow-up of 3.0+/3.2 years has been mentioned, which means that some patients had a very short follow-up and some patients had only two HbA1c readings. Also, since data is not normally distributed, better to present it as median and range? 

Response from authors: In this study, we used HbA1c values to estimate glycemic variability, As HbA1c reflect the glycemic control during the last two to three months, and that we in addition had a follow-up of 3.0 ± 3.2 years, we believe that 'long-term glycemic variability' is adequate in the title. In this study, the median HbA1c was 6.6% and the range was between 4.2-19.7% (Material and Methods, first paragraph, fifth sentence). 

2. Kindly mention the PD regimes in the population studied and adequacy of PD (if possible): 

Response from authors: A. In a large number of patients, PD regimes were changed during the follow-up period. Furthermore, in a significant number of patients, PD regimens were even changed more than one time. It is therefore very difficult to present data on the specific PD regimes, which is the reason why we did not include such data in the manuscript. However, at baseline, 72% of the patients received treatment with CAPD regime and 28% was on APD, which is now stated in the manuscript (Material and Methods, first paragraph, last sentence). B. We do not have data on dialysis adequacy of PD. The study limitations are now described more extensively in the second last paragraph of Discussion. 

3. Separating the patients in standard five quintiles may be preferable?. 

Response from authors: We have also analyzed HbA1c variability according to five quintiles. This analysis showed that the last quintile (CV>3.2) was associated with higher risk of mortality, HR=3.2 compared to first quantile (CV<0.67) as reference category. However, our aim was to evaluate whether very low and very high degree of HbA1c variability are related to increased risk of mortality. As a result, we believe that presenting Hba1c variability in seven categories is relevant.

4. Would like to know the mortality in a similar group of non-diabetics on PD in the Registry: Response from authors: Unfortunately, we have no data on non -diabetic patients.

Reviewer 2

1. This seems to be a retrospective observational study retrieving data from a National (Swedish) Registry. The aim of this study was to investigate the association between HbA1c variability, as a measure of long-term glycemic variability, and the risk of all -cause mortality in diabetic patients with PD. 

Response from authors. We thank the Reviewer for this description of the design and aims of our study.

2. In the first paragraph of the Methods section, the authors state that 795 diabetic PD patients were followed up between 2008-2018, but 48 were excluded as they had been on PD for less than three months. Of the remaining patients, 325 had at least two values of HbA1c and were included in the HbA1c variability analyses with a mean follow-up period of 3.0 ± 3.2 years. Would you confirm that 322 were excluded for not having, at least, two values of HbA1c ? Therefore, a total of 370 patients (roughly 46 % of all diabetic PD patients on the Registry) were excluded for either being on PD for less than three months or not having two values or more of HbA1c during the follow-up period (3.0 ± 3.2 years). 

 Response from authors: The Reviewer is correct. Thus, the majority of the patients that were excluded had only one measurement of HbA1c and could therefore not be included in the analyses of HbA1c variability. We believe that this is clearly described in the manuscript in the first paragraph of Material and Methods. 

3. Certainly, all 795 diabetic PD patients followed up between 2008-2018 must have had more than two values of HbA1c analyzed during their follow-up period, but most likely not captured for the Registry. Can this be considered a bias ? a limitation of the study ? and what about the variability of the results measured in different laboratories around Sweden ? were all PD patients in Sweden evaluated for their HbA1c levels with the same lab method ? It is 

Response from authors: A few dialysis centers have not regularly reported their HbA1c values to the Swedish Renal Register (SRR). In this study, we only included the patients with at least two HbA1c values. It is now acknowledged in Discussion (second last paragraph, last sentence) that it is a study limitation that only 325 of the totally 795 patients were included in the present study. Finally, all laboratories in Sweden use the same method to analyze HbA1c. Thus, assay variability was likely of relatively small importance in terms of the accuracy of the data presented in the manuscript. Finally, in the new Table 2, the distribution of number of HbA1c measurements per patient according to classes of HbA1c variability is presented. As seen in the new Table 2, the number of HbA1c measurements did not appear to be associated with the level of HbA1c variability.

4. It possible to know how many of the 322 patients excluded for not having, at least two values of HbA1c, had no (0) values of HbA1c ? 

Response from authors: All the totally 795 patients had at least one HbA1c value. However, to evaluate the association between HbA1c variability and mortality, we only included the patients with at least three months on PD treatment and two or more values of HbA1c (n=325). 

5. In the Introduction the authors write ……… Earlier studies evaluated the association between the mean value of HbA1c and the risk of cardiovascular events or mortality in patients with DM and PD [16-18]. However, the impact of the glucose load from the PD dialysate on glycemic variability has not been previously investigated in diabetic patients. In order to evaluate the impact of glucose load from the PD fluid on glycemic variability, it is fundamental to have important clinical information which may significantly impact on HbA1c variability , the most important being:

a. Data on which modality (APD or CAPD) and the daily prescription for each patient. Were there switches from APD to CAPD or from CAPD to APD in any of the patients included, during the observation period ? 

Response from authors: At baseline, among the 325 patients included in the present study, 235 (72%) were on CAPD and 90 (28%) were on APD. Switches were common during the follow-up period and were mostly temporary. At the end of study, 194 (60%) of the patients had CAPD and 131 (40%) of the patients were on APD . 

b. CAPD daily dialysis prescription: number of exchanges, volume per exchange, glucose concentration in each glucose-based PD solution, use of the non-glucose based PD solutions icodextrin (Extraneal) and or aminoacids solution (Nutrineal)? 

Response from authors: We do not have data on dialysate solutions, which is now acknowledged as a study limitation (Discussion, second last paragraph, third sentence). 

c. APD daily dialysis prescription: number of exchanges, volume per exchange, total volume used, glucose concentration in each glucose-based PD solution, use of the non-glucose based PD solutions icodextrin (Extraneal) and or aminoacids solution (Nutrineal).

Response from authors: These data were not available, which is now acknowledged in Discussion, second last paragraph, third sentence.

c. Daily insulin prescription in case of insulin-dependent diabetics and other drugs for non-insulin dependent diabetics. How many insulin-dependent and how many non-insulin dependent patients were on the study ? 

Response from authors: All patients were treated with insulin. Data on the daily insulin doses were not available (Discussion, second last paragraph, second last sentence).

d. How was the peritoneal membrane transport characteristics distribution among the CAPD and APD patients ? High transporters usually absorb more glucose and this can impact on morbidity and mortality

Response from authors: We used data from Swedish Renal register (SRR), in which data on type of peritoneal membrane transport are not included. This has been included in the manuscript as a study limitation (Discussion, second last paragraph, third sentence). 

Information on blood pressure (percentage of hypertensive patients) and/or fluid overload status. 

Response from authors: In this study, 292 (90%) of the patients had at least one medication for treatment of high blood pressure. We defined hypertension as systolic blood pressure (SBT) > 140 mmHg or diastolic blood pressure (DBP) > 90 mmHg. Using this definition, 146 (47%) of the patients had hypertension (Clinical classification, second paragraph).

e. Use of BRA, ACEi, aspirin, beta-blockers 

Response from authors: Unfortunately, we do not have data about these medications.

f. Presence of co-morbidities ( Charlson index ?).

Response from authors: In this study, we only have data on cardiovascular disease and previous malignancy. Cardiovascular disease included as coronary heart disease (CHD), stroke or peripheral arterial disease (PAD).

g. Information on peritonitis rates, Kt/V, residual renal function.

Response from authors: We have no data on dialysis effectivity and Kt/V, which is stated in Discussion, second last paragraph, third sentence.

6. In the introduction the Authors wrote…. However, the impact of the glucose load from the PD dialysate on glycemic variability has not been previously investigated in diabetic patients. Information above described from a to h should be provided in order to corroborate this assumption. ??

Response from authors: In response to this comment, we have slightly modified this statement in Introduction to: “However, the impact of the glucose load from the PD dialysate on glycemic variability in relation to mortality has previously not been investigated in diabetic patients.” To our knowledge, this has previously not been investigated in diabetic patients.

7. The authors conclude….” In conclusion, to our knowledge, the present study is the first to evaluate the association between glycemic variability and the risk of all-cause mortality in diabetic patients receiving PD treatment. In our diabetic patients with maintenance PD, high HbA1c variability, as a measure of long-term glycemic control, was significantly associated with increased risk of all-cause mortality. Therefore, higher magnitudes of glycemic fluctuations, which might be caused by radical changes in dialysis regimes or peritonitis, are associated with higher risk of mortality in this group of patients. Further studies are needed to evaluate whether improved clinical care can reduce glycemic variability as well as the high mortality seen in patients with DM and PD. “Are they sure that “they are the first to evaluate the association between glycemic variability and the risk of all-cause mortality in diabetic patients receiving PD treatment.” ? 

Response from authors: We have evaluated previous studies and we did not find any study that assessed the aim of our study ``association between HbA1c variability and the risk of mortality in patients with diabetes mellitus on PD´´. Most of the earlier studies in PD populations examined the association between the mean value of HbA1c and the risk of mortality or cardiovascular events. However, we have slightly modified the last paragraph of Discussion in response to this comment.

8. When the authors write “In our diabetic patients with maintenance PD…”, isn’t it more appropriate to write “In the SRR diabetic patients group with maintenance PD…” ? 

Response from authors: In response to this comment from the Reviewer, we have changed this sentence to ´´In this observational study using data from the SRR of diabetic patients with maintenance PD, high HbA1c variability, as a measure of long-term glycemic control, was significantly associated with increased risk of all-cause mortality´´.

9. I cannot see support to the following sentence “Therefore, higher magnitudes of glycemic fluctuations, which might be caused by radical changes in dialysis regimes or peritonitis, are associated with higher risk of mortality in this group of patients “ as there is no information or data on dialysis prescriptions or peritonitis rates analysed in this study as well as of a definition of the word “radical” in the context of “radical changes in dialysis regimes or peritonitis “. In order to address this association (changes in dialysis prescriptions or peritonitis) with mortality, what is needed to investigate is primarily not the glycemic variability (a consequence of glucose exposure), but the changes in the daily glucose exposure/load and its clinical impact on different sub-groups of diabetic PD patients. In diabetic PD patients, higher HbA1c levels may indicate greater cumulative peritoneal glucose exposure with its attendant damage to the peritoneal membrane. 

Response from authors: The main aim of this study was to evaluate the association between HbA1c variability, as a measure of long-time glycemic control, and the risk of mortality. However, as mentioned in manuscript, there is a need to examine the impact of the changes in the daily glucose exposure/load by dialysis regimes, type peritoneal membrane and even peritonitis, to determine their importance for the association between glycemic variability and mortality in PD populations in future studies. 

10. The authors wrote as last sentence in the conclusion…. “Further studies are needed to evaluate whether improved clinical care can reduce glycemic variability as well as the high mortality seen in patients with DM and PD. “. I would challenge this conclusion using as basis, two not recent published randomized trials (Paniagua et al, de Moraes et al), already showing improved clinical care by improving metabolic control, decreasing glucose load and exposure as well as optimizing fluid management in PD patients (both diabetics and non-diabetics). I suggest reading the RCT CAPD study by Ramon Paniagua et al. and the RCT APD study by Thyago de Moraes et al. 

Paniagua R, Ventura MD, Avila-Diaz M et al. Icodextrin improves metabolic and fluid management in high and high-average transport diabetic patients. Perit Dial Int 2009; 29: 422–432 de Moraes T.P., Andreoli M.C., Canziani M.E. et al. Icodextrin reduces insulin resistance in non-diabetic patients undergoing automated peritoneal dialysis: results of a randomized controlled trial (STARCH). Nephrol Dial Transplant. 2015; 30: 1905-1911. also suggest reading the RCT Impendia study by Li PK et al

In the IMPENDIA study, the primary endpoint was change in glycated hemoglobin from baseline. Mean glycated hemoglobin at baseline was similar in both groups. During the six months of the study, in the intention-to-treat population, the mean glycated hemoglobin profile improved in the intervention group but remained unchanged in the control group (0.5% difference between groups; 95% confidence interval, 0.1% to 0.8%; P=0.006).

Li PK, Culleton BF, Ariza A et al. Randomized, controlled trial of glucose sparing peritoneal dialysis in diabetic patients. J Am Soc Nephrol 2013; 24: 1889–1900

As well as reading the paper by McIntyre et al on glycemic control in diabetic CAPD patients assessed by CGMS

A practical approach to reduce disturbances of the carbohydrate metabolism in PD patients is the reduction of glucose exposure by also prescribing glucose-sparing solutions. In a study involving eight diabetic CAPD patients, replacement of a glucose-based regimen with a Physioneal-Extraneal-Nutrineal regimen was associated with a reduction in the 24-hour variability of glucose concentrations as measured by a subcutaneous probe in the interstitial fluid of the abdominal wall.

Marshall J, Jennings P, Scott A, Fluck RJ, McIntyre CW: Glycemic control in diabetic CAPD patients assessed by continuous glucose monitoring system (CGMS). Kidney Int 64: 1480–1486, 2003

 Response from authors: Many thanks for very interesting articles, which we have read carefully and added as references in our manuscript. The results of these studies show that non-glucose-containing dialysis fluid was associated with better metabolic control. The result of our study indicated that higher HbA1c variability, as measure of long-term glycemic control, was associated with increased risk of mortality. In the manuscript, we suggest that the use of non-glucose-containing dialysis fluid like Icodextrin could possibly be related to lower variability of blood glucose as well as a decrease in the risk of mortality among PD patients with diabetes. This needs to be assessed in further observational and RCT studies. 

11. Lastly, it should be taken into account that in type 1 and 2 diabetics (not in dialysis), HbA1c is not a good predictor of cardiovascular disease (CVD), whereas insulin resistance is predictive of CVD and indeed may be the most important single cause of coronary artery disease (see for example: Home P. Diabetes Care 2019; 42:1615-23; Adeva-Andany MM et al. Diabetes Metab Syndr 2019; 13:1449-55; Shahim B. et al. Diabetes Care 2017:40:1233-40; Eddy D. et al. Diabetes Care 2009; 32; 361-66; Orchard TJ et al. 2003; 26:1374-79). In addition, since patients in dialysis are often affected by subclinical/clinical anemia, which reduces red blood cell survival and hence Hb glycosylation, this introduces an additional bias in the interpretation of HbA1c variability. At this regard, it should be kept in mind that the simple evaluation in dialysis patients of total Hb levels would be of little help in figuring out potential differences in the glycosylation rate as total Hb levels. Indeed, the latter (Hb levels) may be easily corrected by Epo treatment, but it doesn’t tell you anything about survival rate of RBCs, one of the major determinant of the extent of HB glycosylation.

Response from authors: As discussed in the manuscript, the use HbA1c as a measure of long-term glycemic control is debated in patients with end stage renal disease (ESRD) and dialysis treatment. However, markers other than HbA1c, like glycated albumin and fructosamine have some major disadvantage in dialysis populations, why we believe that HbA1c still is the best marker of long-term glycemic control.

In my opinion the paper needs a major revision. If the Authors do not have the clinical data necessary to make the association in the title robust and as a reflection of the clinical reality, you may shorten the article with a statistical description of what you see from the data available. I am not an expert in Statistics, but as a clinician, I consider the clinical data I described above of utmost importance in order to keep the paper in the present format. So I really hope you have the data I am suggesting to add to the paper.

---

## [Editor Report · Decision Letter 1]

10 Jan 2022

LONG-TERM GLYCEMIC VARIABILITY AND THE RISK OF MORTALITY IN DIABETIC PATIENTS RECEIVING PERITONEAL DIALYSIS

PONE-D-21-18957R1

Dear Dr. Peters,

We’re pleased to inform you that your manuscript has been judged scientifically suitable for publication and will be formally accepted for publication once it meets all outstanding technical requirements.

Kind regards,

Vivekanand Jha

Academic Editor

PLOS ONE
---

## [Editor Report · Acceptance letter]

17 Jan 2022

PONE-D-21-18957R1 

LONG-TERM GLYCEMIC VARIABILITY AND THE RISK OF MORTALITY IN DIABETIC PATIENTS RECEIVING PERITONEAL DIALYSIS 

Dear Dr. Peters:

I'm pleased to inform you that your manuscript has been deemed suitable for publication in PLOS ONE. Congratulations! Your manuscript is now with our production department. 

Kind regards, 

on behalf of

Prof Vivekanand Jha 

Academic Editor

PLOS ONE